# Mechanical and Microstructural Investigations of the Laser Welding of Different Zinc-Coated Steels

**Eva Zdravecká [1],\* and Ján Slota [2]**

[1] Department of Mechanical Technology and Materials, Faculty of Mechanical Engineering, Technical University of Košice, Mäsiarska 74, 040 01 Košice, Slovakia

[2] Department of Computer Support of Technology, Faculty of Mechanical Engineering, Technical University of Košice, Mäsiarska 74, 040 01 Košice, Slovakia; jan.slota@tuke.sk

\* Correspondence: eva.zdravecka@tuke.sk; Tel.: +421-55-602-3516

**Abstract:** Tailor welded blanks (TWB) represent an anisotropic and non-homogenous material. The knowledge of the mechanical properties and microstructure of the fusion zone and heat-affected zone (HAZ) obtained with laser welding is essential to ensure the reliability of the process. In this paper, laser-welded hot-dip Zn-coated low carbon microalloyed steels with different thickness and mechanical properties were used. The mechanical properties of the laser-welded blanks were determined by tensile tests and formability by Erichsen cupping tests. In addition, the pore formation during the laser welding process was analyzed. The microstructural analysis confirmed the formation of the favorable structure of the weld metal and the heat-affected zone without the presence of martensite. The obtained results showed that it is possible to produce TWBs with suitable mechanical properties by laser welding.

**Keywords:** tailor welded blanks; laser welding; Zn-coated low carbon microalloyed steels; microstructure; pores

---

## 1. Introduction

Laser beam welding is very flexible because welds can be made in continuous and complex shapes, can be performed at relatively high travel welding speeds, and can achieve deeply penetrated welds. Tailor welded blanks (TWB) are blanks that have been manufactured from sheets with similar or different thicknesses by a welding process. The differences in the material within a TWB can be in the thickness, grade, or coating of the material, e.g., galvanized versus ungalvanized. Weight and production cost-reducing automotive body design are achieved by using laser butt welded semi products such as tailored blanks made of steel or combined materials. The laser is preferred for welding owing to the high speed of the process, low distortion due to the small heat-affected zone, the manufacturing flexibility, and the ease of automation [1–5].

Designers are able to tailor the location in the blank where specific material properties are desired when creating a TWB. This trend of welding and forming of sheet-metal pieces allows notable flexibility in product design, structural stiffness, and crash behavior (crashworthiness).

The laser-welding of zinc-coated steel sheets is often used in the automotive industry. A major problem that arises when welding these materials is associated with the vaporization temperature of zinc (906 °C), which is much lower than the melting temperature of steel (1530 °C). The laser-welding of galvanized steel sheets in the thickness range of 0.6–1.5 mm is largely performed in the automotive industry for tailored blank welding applications [6–9].

Low carbon microalloyed steel sheets have long been a commonly-used material in consumer industries because it can be stamped into inexpensive parts with complicated shapes at very high

production rates [10]. Interstitial-free (IF) steel sheets are the most frequently used sheet material for complex automotive applications due to their superior formability [11–15].

Previous studies and research have identified the factors that have an important effect on the strength of the laser beam-welded joints. These are the laser power (P), welding speed (S), and focal position (F), which were considered to be the factors influencing the Erichsen cupping test results and the microstructure of specimens [1,2,16].

Some important evaluated factors of TWBs are material property changes to welds made of different combinations of microalloyed steel sheets and changes of non-uniform deformation because of the differences in thickness, properties, and surface characteristics with respect to the application of the load.

This investigation addressed three types of galvanized automotive steel sheets, namely:

1. DX54D+Z (EN10346:2015)—interstitial-free (IF) steel sheets appropriate for galvanizing and annealing to produce specialized steel sheets required for automotive body manufacturing. IF steel sheets are free from carbide precipitates at the grain boundaries.

2. DX53D+Z (EN10346:2015)—low carbon steel sheets, where the microstructure is ferrite–pearlite with polyedric ferrite grains and sporadically precipitated deformed pearlite. Hot-dip galvanized sheets made of drawing grades are suitable for cold forming and deep-drawing. The sheets are used for the production of automobile parts, appliances, in the building industry, and for production of profiles, corrugated sheets, roof coverings, and engineering.

3. ZStE260Z (SEW 093/87)—the ferritic microstructure cementite is precipitated in ferrite grains. The structure is fine-grained.

The present work focuses on the characterization of the butt joint of low carbon galvanized steel sheets finished by laser welding. The influence of the laser welding on the microstructure, phase transformations, hardness, mechanical strength, and work hardening behavior is also investigated. The novelty of the paper concerns the evaluation of the special combinations of Zn-coated steel sheets (TWBs). The combination of materials labeled TWB1 and TWB2 has been requested by the manufacturer of tested materials for intended use in laser welding and has potential for applications in the automotive industry.

## 2. Materials and Methods

### 2.1. Materials

The experiments were conducted on zinc-coated steels. Three types of galvanized steel sheets in the thickness range of 0.8 to 1.75 mm with differential mechanical properties were used for tailored blank welding (TWB) applications and marked as TWB1 and TWB2 (see Tables 1 and 2). For the experiment, combinations of hot-dip zinc coated low carbon steel sheet DX53D+Z (deep drawing quality sheet steel for cold forming according to EN 10327) with DX54D+Z interstitial-free (IF) steel sheets marked as TWB1 were used. The second combination, marked as TWB2, consisted of low carbon microalloyed steel sheets DX53D+Z in combination with ZStE260Z. The chemical compositions and thicknesses of TWB1 and TWB2 are shown in Tables 1 and 2.

**Table 1.** Chemical composition and thickness of TWB1.

| Material | Thickness (mm) | Chemical Composition TWB1 (%) | | | | | | | | |
|---|---|---|---|---|---|---|---|---|---|---|
| | | $C_{max}$ | $Mn_{max}$ | $P_{max}$ | $S_{max}$ | $Si_{max}$ | Al | Ti | $Nb_{max}$ | $N_{max}$ |
| DX54D+Z (IF) | 0.80 | 0.015 | 0.20 | 0.015 | 0.015 | – | 0.02 | 0.06–0.14 | – | 0.006 |
| DX53D+Z | 1.00 | 0.04 | 0.20 | 0.015 | 0.012 | 0.01 | 0.03–0.06 | – | – | 0.006 |

**Table 2.** Chemical composition and thickness of TWB2.

| Material | Thickness (mm) | Chemical Composition TWB2 (%) | | | | | | | | |
|---|---|---|---|---|---|---|---|---|---|---|
| | | $C_{max}$ | $Mn_{max}$ | $P_{max}$ | $S_{max}$ | $Si_{max}$ | Al | Ti | $Nb_{max}$ | $N_{max}$ |
| DX53D+Z | 1.75 | 0.04 | 0.20 | 0.015 | 0.012 | 0.01 | 0.03–0.06 | – | – | 0.006 |
| ZStE260Z | 1.30 | 0.10 | 0.60 | 0.025 | 0.008 | 0.04 | 0.015 | 0.04 | 0.02–0.035 | – |

## 2.2. Process Parameters of Laser Welding

A $CO_2$ laser with a nominal power of 5 kW was used for laser welding. The $CO_2$ laser parameters were: wavelength—10.6 mm; power range—500 W to 5000 W; output stability—+2%; diameter of output beam—13 mm; mode—TEMoo; divergence—1.5 mrad; point stability—3.35 mrad; pulsing—up to 1 kHz; and horizontal beam polarization. In order to avoid accidental misalignment or movement during laser welding, a welding sample fixture was used to ensure stable and consistent experimental tests. Welding was carried out without gaps [17,18]. During welding, gas argon was used as the shielding gas. The butt weld was processed with full penetration on the sheet steels of uniform thickness. The welding parameters that were set are shown in Table 3.

**Table 3.** Welding parameters.

| LBW Parameters | TWB1 | TWB2 |
|---|---|---|
| Laser power (P) (kW) | 2.5 | 2.9 |
| Welding speed (s) (mm·s$^{-1}$) | 45 | 45 |
| Shielding gas pressure (P) (MPa) | 0.2 Ar | 0.2 Ar |
| Focal position (F) (mm) | 0 | 0 |

## 2.3. Metallography and Hardness Testing

Samples for metallographic examination were cut from each of the TWBs and prepared following standard procedures. Polished specimens were etched using 3% Nital and the microstructures viewed under the Carl Zeiss Jena NEOPHOT 32 optical microscope. These analyses were focused to study the microstructure in the fusion zone (FZ), heat-affected zone (HAZ), and base metal (BM).

One of the most-commonly used techniques to determine the mechanical properties in the welding zone is microhardness measurement. The microhardness of the base materials and welding joints were analyzed by SHIMADZU–DUH 202, Japan, Indenter VICKERS (diamant), using a load of 98 mN (10 gf). The Vickers hardness test was performed according to ISO 6507. The hardness was measured 3–5 times in each area, and the average value was taken as the test result.

## 2.4. Tensile Tests and Formability

The dimensions of the tailor welded sheet metals were 250 mm × 250 mm. The shaped specimens were cut out perpendicular to the weld line from welded blanks. All the specimens were machined carefully with the centerlines of the weld zone and the specimen being coincident, as illustrated in Figure 1. The aim of the tensile tests was to evaluate the strength and plasticity of welding joints and/or examine the influence of welding defects on the joint performance. The tensile test was performed on a universal testing machine ZWICK 50 according to EN 895 and EN ISO 6892-1:2016. Standard tensile characteristics include yield stress (YS), ultimate tensile strength (UTS), and percent over elongation at break (ductility). The YS, UTS, and elongation to fracture of the BM as well as TWB were evaluated [19–23]. In the calculation of the stress-strain diagram of the TWB samples, the thickness of the thinner sheet was considered as the initial thickness. Cracking of the sample was not observed in the FZ or HAZ, but on the side of the material with lower mechanical properties or lower thickness [24].

The Erichsen cupping test, one of the conventional formability test methods, was used to evaluate the welded blank formability. Specimens for the Erichsen cupping test were prepared according to ISO 20482.

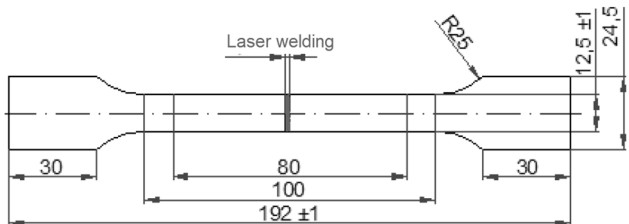

**Figure 1.** Shape and dimensions of the welded specimen (tension perpendicular to weld plane) (Unit: mm).

## 3. Results and Discussion

### 3.1. Microstructure Characterization of the Laser Welding Joints

The mechanical properties obtained in different welding zones are a function of the microstructure. To understand the welding behavior, it is helpful to examine the microstructure of the base materials and welded joints [25,26]. Analyses of the microstructure in the fusion zone (FZ), heat-affected zone (HAZ), and base metal (BM) for all specimens were performed. An overall cross-section photo of the microstructure of laser-welded steel sheets (DX54D+Z—0.8 mm/DX53D+Z—1.0 mm) marked as TWB1 is seen in Figure 2. The overall top width of the weld was approx. 0.7 mm, the bottom of the weld was 0.8 mm, the HAZ (DX54D+Z) was approx. 0.3 mm, and width of the HAZ (DX53D+Z) was approx. 0.4 mm (see Figure 2).

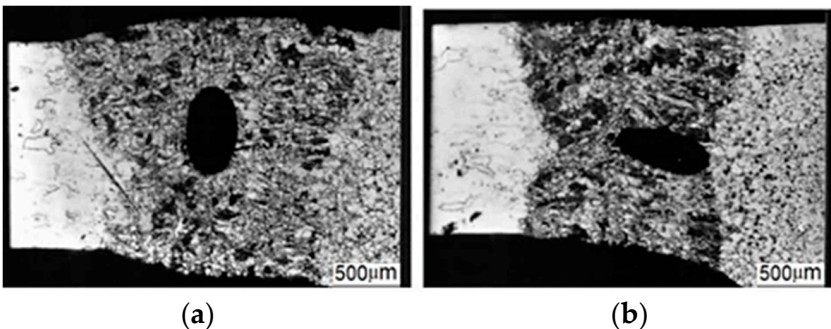

| (a) | (b) |

**Figure 2.** Micrographs of the TWB1 laser-welded specimens: (**a**) the pore in central area of FZ, (**b**) the pore near HAZ.

The transition zone FZ/BM was smooth, without micropores and notches. The weld metal contains multiple pores or cavities. The top and bottom of the weld have a smooth transition between sheets of different thicknesses. More detailed microstructures of the TWB1 laser-welded joint are shown in Figure 3a–e.

Figure 3a FZ: the ferritic-bainite microstructure, pearlite sporadically precipitated as untransformed pearlite (the presence of the cavity was observed).

Figure 3b HAZ: DX54D+Z (IF); ferritic-cementite microstructure is in the area of high temperature overheating of the base material, with a singularly formed bainite, and in the $A_{C1}–A_{C3}$ temperature range the ferritic-cementite microstructure is with uniquely precipitated carbides.

Figure 3c HAZ: DX53D+Z; the ferritic-pearlitic microstructure in the area of high overheating of the base material with a singularly formed bainite; in the $A_{C1}–A_{C3}$ temperature range the microstructure is ferritic-pearlitic with deformed pearlite, and ferrite has enlarged grains.

Figure 3d BM: DX54D+Z; the ferritic microstructure with a uniquely precipitated cementite.

Figure 3e BM: DX53D+Z; ferritic–pearlite microstructure with polyedric (quasi-equiaxed) ferrite grains and sporadically precipitated deformed pearlite.

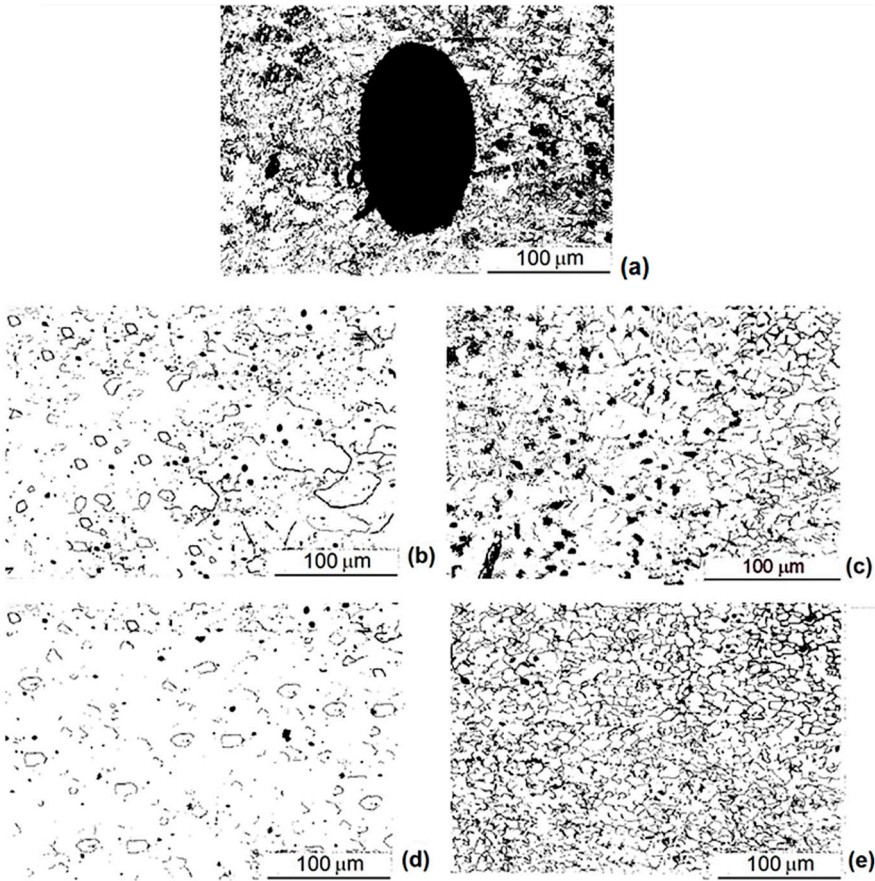

**Figure 3.** Microstructure of TWB1. (**a**) Fusion zone (FZ) (DX54D+Z/DX53D+Z); (**b**) Heat-affected zone (HAZ) (DX54D+Z); (**c**) HAZ (DX53D+Z); (**d**) Base metal (BM) (DX54D+Z); (**e**) BM (DX53D+Z).

Similar analyses for the TWB2 laser-welded specimens were performed. A photo of the microstructure of laser-welded steel sheets (ZStE260Z, 1.3 mm/DX53D+Z, 1.75 mm) marked as TWB2 is seen in Figure 4. The overall width of the top of the weld was approx. 1.0 mm, the bottom of the weld was 0.6 mm, and width of the HAZ (DX53D+Z) was approx. 0.6 mm, and the HAZ (ZStE260Z) was 0.6 mm (See in Figure 4).

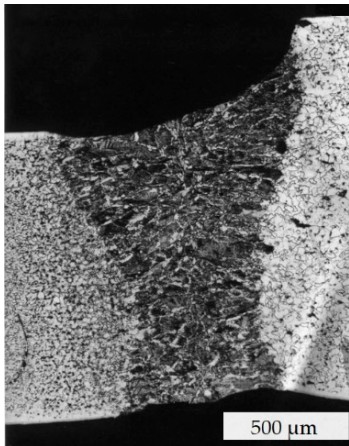

**Figure 4.** Micrograph of the TWB2 specimen.

The microstructure of the TWB2 laser-welded joint are shown in Figure 5a–e.

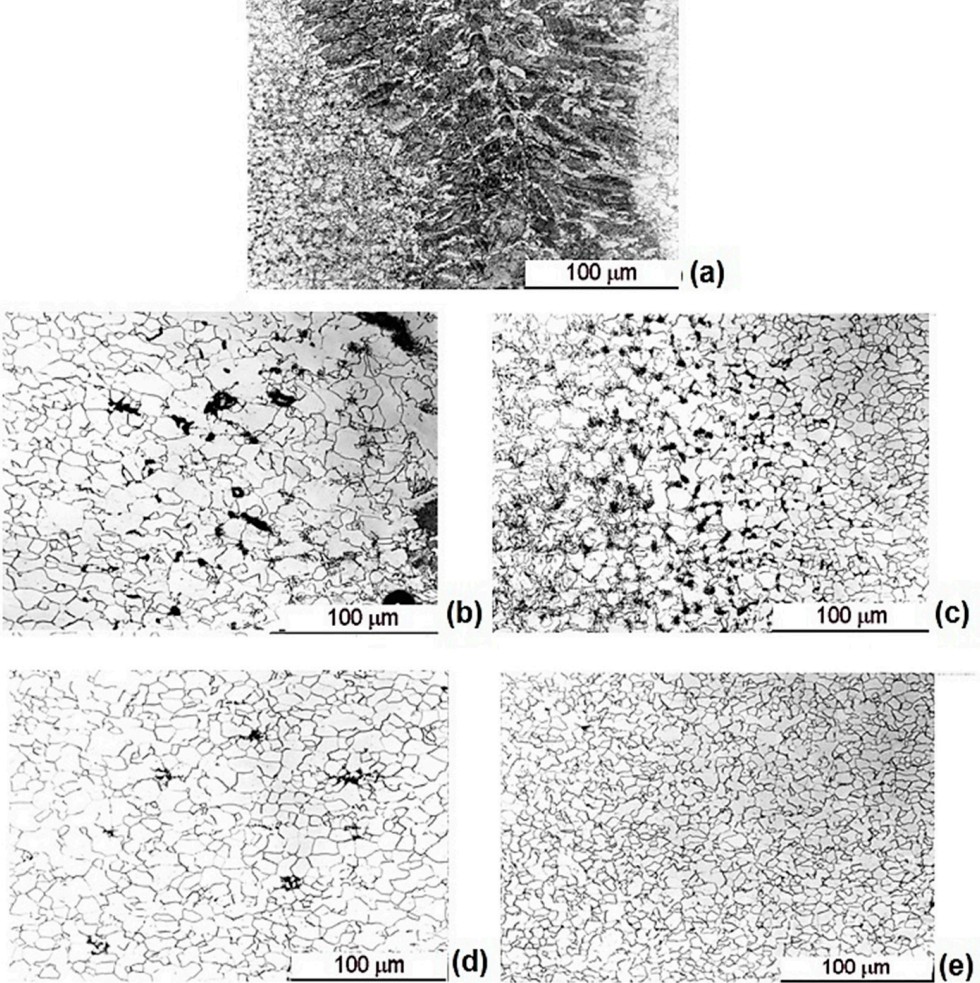

**Figure 5.** Microstructure of TWB2. (**a**) FZ (DX53D+Z/ZStE260Z); (**b**) HAZ (DX53D+Z); (**c**) HAZ (ZStE260Z); (**d**) BM (DX53D+Z); (**e**) BM (ZStE260Z).

Figure 5a FM: Weld metal; the bainitic-ferritic microstructure with acicular ferrite.

Figure 5b HAZ: DX53D; the ferritic-pearlite microstructure in the area of high temperature overheating of the base material with a singularly precipitated bainite; in the $A_{C1}$–$A_{C3}$ temperature range is a ferritic-pearlite microstructure with deformed pearlite and enlarged ferrite grains.

Figure 5c HAZ: ZStE260Z; the ferritic-bainitic-pearlitic microstructure in the area of high temperature overheating of the base material; in the $A_{C1}$–$A_{C3}$ temperature range is a ferritic-pearlite microstructure with enlarged ferrite grains and deformed pearlite.

Figure 5d BM: DX53D+Z; the ferritic-pearlite microstructure with polyedric ferrite grains and sporadically precipitated deformed pearlite.

Figure 5e BM: ZStE260Z; the ferritic microstructure, cementite is precipitated in ferrite grains. The structure is fine-grained.

The microstructure of TWB1 and TWB2 did not show any evidence of the excessive root penetration of the welds. In the HAZ, the grain thickness was slightly increased. For the microalloyed steel, pearlite and bainite were precipitated in the HAZ, and bainite was precipitated in the low carbon steel.

## 3.2. Analysis of the Microhardness

The microstructure hardness was measured in the base metal (BM), heat-affected zone (HAZ), and fusion zone (FZ). The microhardness distribution in the base metal, heat-affected zone, and fusion zone were measured for TWB1a and TWB2. In Figure 6 (TWB1) and Figure 7 (TWB2) marks represent

the hardness testing position in the base metal zone, heat-affected zone, and fusion zone, respectively. The average values of HV0.5 microhardness and standard deviation (Stdev) for each material are shown in Figure 6.

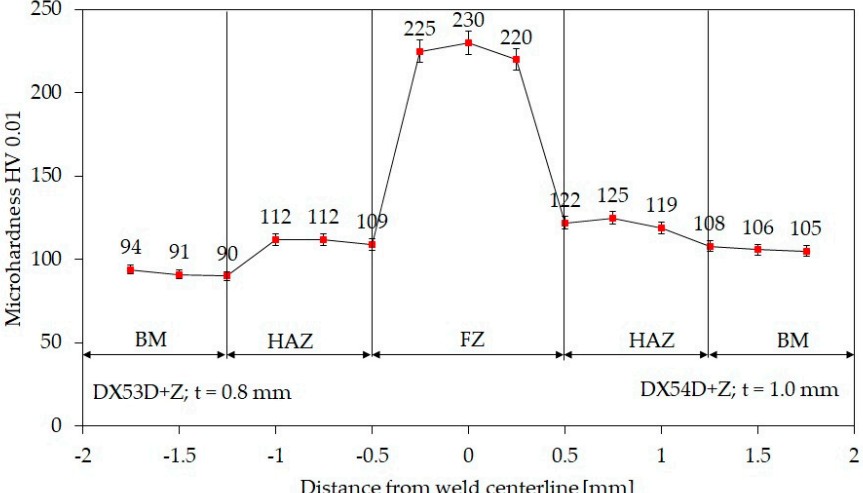

**Figure 6.** Microhardness distribution of TWB1.

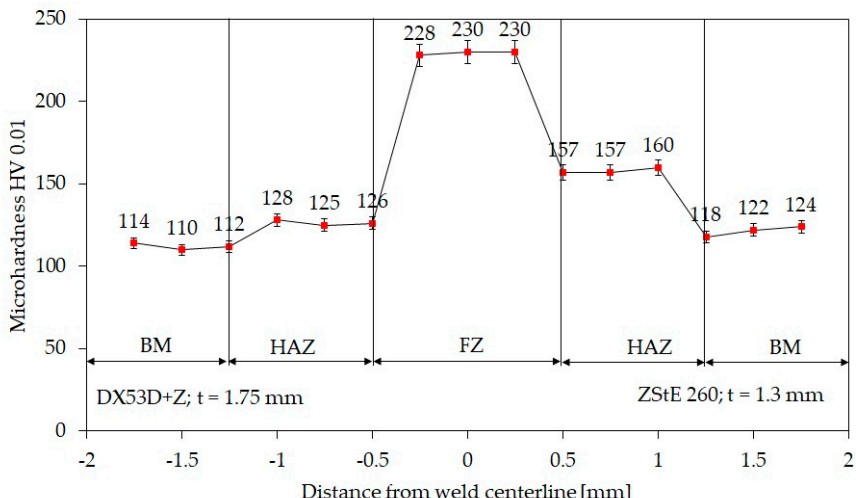

**Figure 7.** Microhardness distribution of TWB2.

The microhardness of the HAZ in DX53D+Z steel increased up to 112 HV and in the DX54Z+Z steel it increased up to 125 HV. The widths of the HAZ and FZ in the welded joints were similar. The higher microhardness of the FZ can be explained by the presence of bainite in the microstructure.

The ZStE260 material contains a larger amount of pearlite and a certain amount of cementite, which explains the higher microhardness in the HAZ up to 160 HV (Figure 7). Figures 6 and 7 show that the microhardness in the FZ of the welded joints was similar.

### 3.3. Results of the Tensile and Formability Tests

One of the aims of a tensile test is to evaluate the strength and plasticity of welding joints and examine the influence of welding defects on the joint performance [27]. Figures 8 and 9 show the results of the mechanical properties of the base materials (thinner blanks) and TWB welded joints.

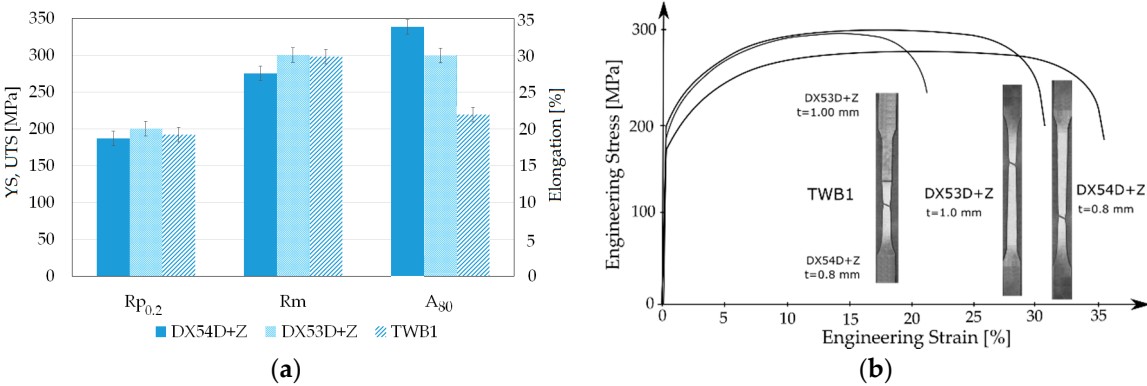

**Figure 8.** (**a**) The yield stress, tensile strength, and elongation of the base materials and TWB1; (**b**) the stress–strain curves of the base materials and TWB1.

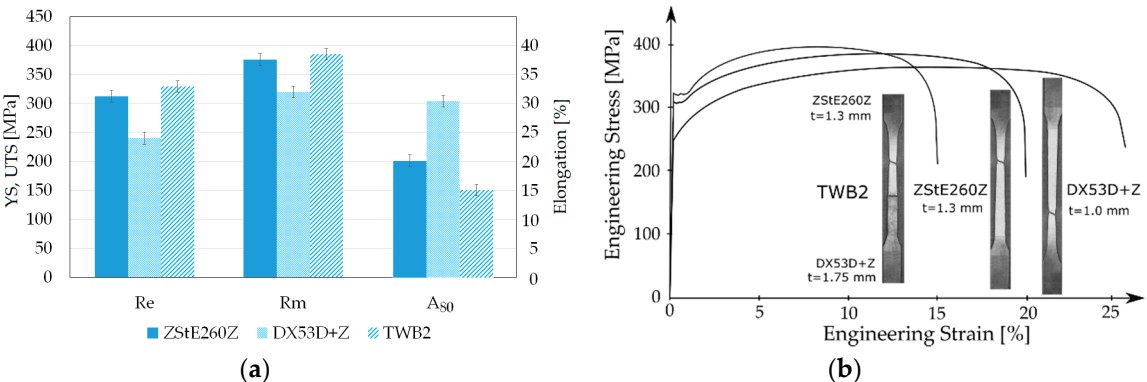

**Figure 9.** (**a**) The yield stress, tensile strength, and elongation of the base materials and TWB2; (**b**) the stress–strain curves of the base materials and TWB2.

The elongation of the welded joints decreased with the increased hardness of the welds. It was observed that the fracture occurred in the thinner/weaker material, but not in the weld, as shown in Figure 10.

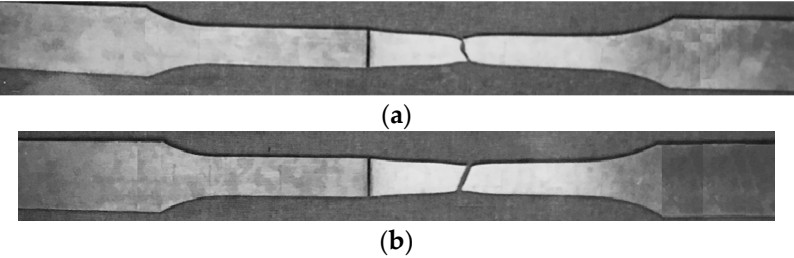

**Figure 10.** The TWB samples after tensile tests: (**a**) TWB1, (**b**) TWB2.

The failure occurred in the thinner sheet metal for both TWBs, which conforms with the literature [28–31]. A number of studies showed that increasing the thickness and/or strength ratios decreases the formability of the TWBs [29–31]. A larger thickness ratio forces more deformation into the weaker material and the strain is concentrated there, which results in premature failure. During deformation, the thinner material undergoes plastic deformation, whereas the thicker material undergoes primarily elastic deformation. An increase in the strength ratios has a similar effect on the failure mode as the thickness ratio, whereby the weaker material deforms more and fails first. In addition, due to non-uniform deformation, the weld line also tends to move towards the thicker/stronger materials.

The obtained results indicated that the microhardness of the weld zone was higher than the microhardness of the base sheets. This shows that the thinner blank is dominant in the overall deformation behavior. The thinner blank is the one that will decide the properties of the whole TWB. The thicker blank behaves as a rigid support to the thinner blank [32,33].

The Erichsen cupping test value of the base materials and TWB1 is shown in Figure 11. From the tested samples, the highest cup depth of 11.35 mm was obtained.

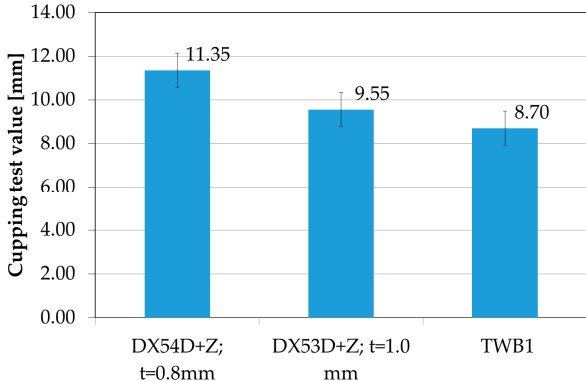

**Figure 11.** The Erichsen cupping test value of the base materials and TWB1.

The crack spread in a circular shape along the weld. On TWB1, the crack was initiated outside of the HAZ, despite internal defects in the weld (Figure 3). Figure 12a shows the initiation of the failure in base material DX54Z+Z and propagation transverse to the weld line into the base metal DX53D+Z. Figure 12b shows a ductile fracture of the base material DX54D. From the measured microhardness values, we can see that the laser welding caused material hardening.

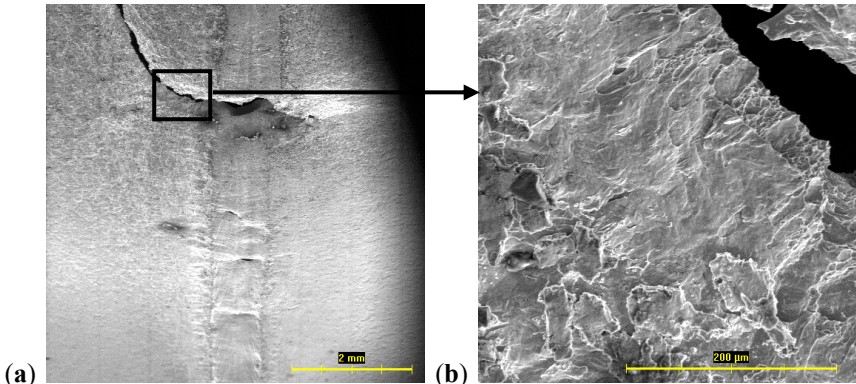

**Figure 12.** Failure of TWB1 after the Erichsen test. (**a**) The initiation and propagation of the failure. (**b**) Fracture surface.

The presence of defects in the welded joint could be affected by the improper machining of the welding edges and degreasing. The concentration of large amounts of energy during laser welding vaporizes the burrs after cutting, as well as impurities and grease on the surface of the sheet. The vapors from zinc coating as well as impurities get into the weld metal and remain blocked due to rapid cooling. The pore formation was identified on the bottom of weld for TWB1 (Figures 12 and 13a,b). The Energy Dispersive X-ray Analysis (EDX) analysis of the pore in TWB1 is shown in Figure 13c.

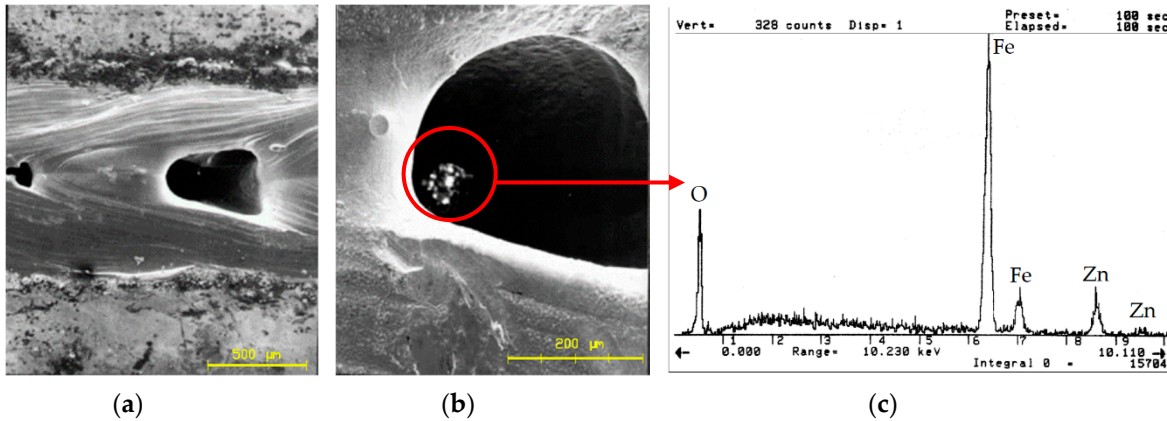

**Figure 13.** Laser weld line on TWB1 with a pore: (**a**) Pore formation in the weld, (**b**) A detailed view of the pore, (**c**) EDX analysis of the pore in the TWB1 sample.

It was found that thin sheets are more sensitive to the welding parameters, surface cleanness, and preparation of edges. The bubbles showed mixed behavior, indicating that the capillary is partly filled with zinc vapor and partly with ambient gas. Figure 14b shows the SEM of the cross–section of the TWB1 weld with the internal pore. Figure 14a,c shows the results of the EDX analysis. The advantage of the EDX analysis, in this case, is the exact identification of the chemical composition of each compound in the pore. This will contribute to the search for the source of the pore formation. The porosity in the TWBs could be due to various causes, e.g., keyhole instability, improper gas shielding, surface contamination, improper edge preparation, and the improper setting of welding parameters. The setting of the laser power affects the evaporation of the substrate and creates a good key effect. This creates an opening that can pump high-pressure zinc gas and reduce the risk of explosion.

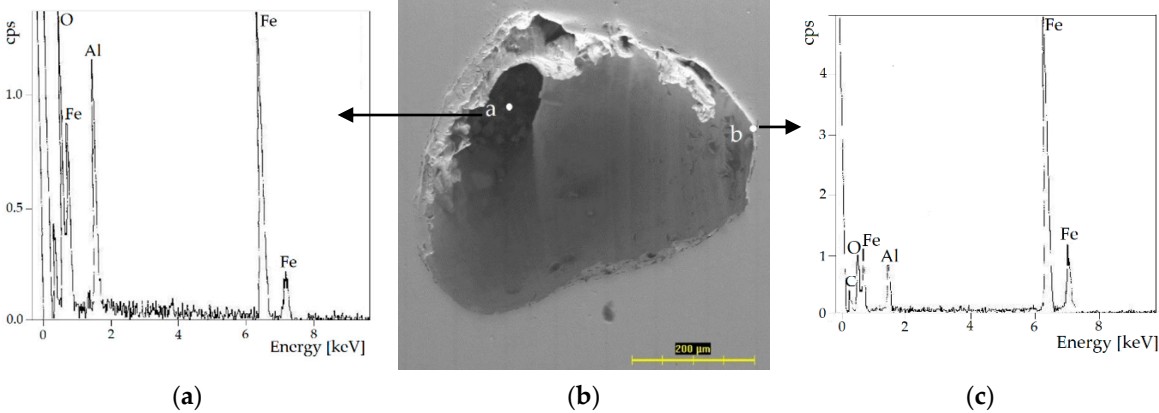

**Figure 14.** Cross-section through the TWB1pore. (**a**) EDX analysis of the pore; (**b**) SEM of the cross-section; (**c**) EDX analysis of the edge of the pore.

The Erichsen cupping test value of the base materials and TWB2 is shown in Figure 15. The base material ZStE260, with a thickness of 1.3 mm, showed a circular-shaped crack. In TWB2 specimens, the crack was of circular shape, and failure occurred a sufficient distance from the weld in the thinner sheet metal.

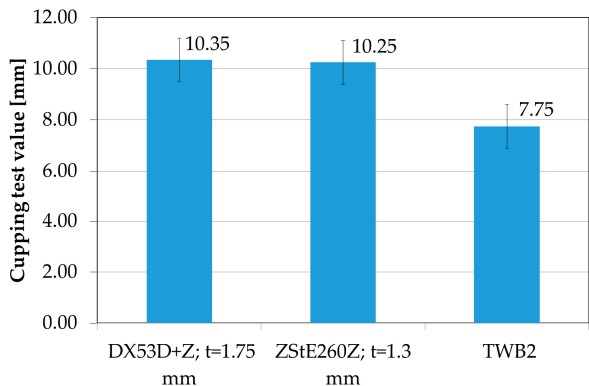

**Figure 15.** The Erichsen cupping test value of the base materials and TWB2.

## 4. Conclusions

The laser beam welding process and its effects of the on the quality of the welded joints were analyzed on microalloyed, high-strength, Zn-coated steels with different thicknesses (0.8 and 1.0 mm/TWB1 and 1.3 and 1. 75 mm/TWB2). The microstructure and mechanical properties of the laser-welded butt joints were investigated. The main results are summarized as follows:

i.    The good weldability of microalloyed high-strength, Zn-coated steels was confirmed.

ii.   In the microhardness test, hardness peaks were found in the weld metal. There was no evidence of martensite in the HAZ or the weld metal. For microalloyed steel, pearlite and bainite were precipitated in the HAZ, and bainite was precipitated in the low carbon steel. In all the TWBs, the FZ was predominantly formed by ferritic structures, with some grains of low-carbon bainite.

iii.  A change in the mechanical properties of the welded joints and the base materials of the TWB was observed. A decrease of the ductility of both the TWB1 and TWB2 can be related to the heat-affected welding zone due to hardening. The thickness ratio and strength ratio had an effect on the failure, whereby the weaker material (low thickness, lower mechanical properties) deformed more and failed first. A larger thickness ratio forced more deformation into the weaker material and the strain was concentrated there, which resulted in premature failure.

iv.   During the Erichsen test for TWB1, a failure was initiated in the DX54Z+Z base material and its propagation was perpendicular to the weld line. In the TWB2 specimens, the crack was of a circular shape, and the failure occurred a sufficient distance from the weld in the thinner sheet metal.

v.    The possible causes of porosity in the laser welds of hot-dip, Zn-coated, low carbon, microalloyed steel sheets were keyhole instability, improper gas shielding, surface contamination, improper edge preparation, and the improper setting of welding parameters.

vi.   The tensile tests showed lower sensitivity to the detected defect bands. The pores did not have a detrimental effect on the tensile properties of the welded joint, which may be due to the high strength of the fusion zone, which effectively protects the defective weld zone. The strong weld retained the defect and prevented it from spreading. The Erichsen test showed higher sensitivity in the presence of pores.

vii.  The preparation of thinner sheets for welding required consistent weld surface finishing as well as sheet metal fitting. It was found that the thin sheets were more sensitive to welding parameters, surface cleanness, and preparation of edges.

Based on the experimental results, it can be stated that laser-welded blanks designed to produce parts exposed to plastic deformations require quality joints that do not complicate the stamping process. Laser welding is both an instrumentally and technically demanding method, so it is suitable for large-scale production. The problems of setting the right welding parameters and reproducible welding positioning affect the final weld quality.

**Author Contributions:** E.Z. conducted methodology, microhardness test, microscopical analyses, and analysis and interpretation of the results, and funding acquisition; J.S. conducted measurements of the tensile test and the Erichsen test, writing—original draft preparation, editing, visualization.

**Funding:** This research was funded by the Grant Agency of the Ministry of Education, Science, Research, and Sport of the Slovak Republic under grant number VEGA 1/0117/15 and VEGA 1/0259/19.

**Acknowledgments:** The authors are grateful for the support of experimental works by the Grant Agency of the Ministry of Education, Science, Research, and Sport of the Slovak Republic for the support of the project VEGA 1/0117/15 and VEGA 1/0259/19.

**Conflicts of Interest:** The authors declare no conflict of interest.

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
