# Peer review of "Mechanical and Microstructural Investigations of the Laser Welding of Different Zinc-Coated Steels"

_metals, doi:10.3390/met9010091_

Round 1
Reviewer 1 Report
The authors put a great effort in the revision of the resubmitted manuscript. I have read the provided answers to my previous review, and the new version of the paper, thorougly. All my suggestions and major concerns about the weak points of the research have been addressed. Also, the English language and clarity of exposition has been strongly improved reaching a more than acceptable level.
In the present form the paper, though not presenting particular novelties in the field (in terms of laser technology used, and weldment characterization methodological approach), nevertheless provides some valuable data and observations on the weldability, characteristics and performance of Co2 laser welded twbs made of the investigated steel alloys. A welcome addition would have been a more local investigation of the weld seam through dedicated techniques, (such as digital image correlation usage during testing, for instance, or other techniques).
In any case, I eventually reccommend pubblication of the manuscript on Metals.
Author Response
Response to Reviewer 1 Comments
Point 1: The authors put a great effort in the revision of the resubmitted manuscript. I have read the provided answers to my previous review, and the new version of the paper, thoroughly. All my suggestions and major concerns about the weak points of the research have been addressed. Also, the English language and clarity of exposition has been strongly improved reaching a more than acceptable level. In the present form the paper, though not presenting particular novelties in the field (in terms of laser technology used, and weldment characterization methodological approach), nevertheless provides some valuable data and observations on the weldability, characteristics and performance of CO2 laser welded TWBs made of the investigated steel alloys. A welcome addition would have been a more local investigation of the weld seam through dedicated techniques, (such as digital image correlation usage during testing, for instance, or other techniques).
In any case, I eventually recommend publication of the manuscript on Metals.
Response 1: Dear reviewer, thank you for the review of revised manuscript and your suggestions, which can help improve our paper.
Using of local digital image correlation technique (DIC) to evaluate the evolution of local deformation usually are used, but we did not use it when measuring. We used the evaluation of the microstructural analysis and microhardness measurement. These measurements also provide information about the course of local deformation. Results show satisfactory of mechanical properties of TWBs.
In the next research, we will certainly use DIC technique. Thank you again.

Reviewer 2 Report
Presented work give interesting results but looks more like a technical report than a scientific paper. I have some comments, mainly related with tensile test results and interpretation.
The use of a stress-strain curve in experiments (figs 8) has no sense as the sample is made of two different materials with two different cross sections. Authors note that they use the smallest cross-section for deducing the stress, but it has no scientific meaning. They must use load-displacement results. Getting area reduction or elongation on this samples has no meaning, well you can measure it but you cannot compare it directly with the base material.
Figure 8 now include the stress-strain curves of base materials, In these cases the thinnest of each welded sample. As previously mentioned, load-displacement graphs have more sense. It will be interesting to add the load-disp curves for both plates on each case, then in a single figure we will have the welded sample, the material A and the material B together.
Figure 15 show some EDAX results but authors do not comment anything about these results on the text. Take advantage of those measures and analyse them.
Conclusion 3 is related with tensile test results. Direct comparison between welded samples and base material is not possible. If you want to conclude something you must make a more intense study deducing mechanical properties of and ideally welded sample, i.e. taking into account both sections, sample geometry, welding location and mechanical properties of welded plates. Without this study conclusion III is not valid.
Author Response
Response to Reviewer 2 Comments
Point 1: Presented work give interesting results but looks more like a technical report than a scientific paper. I have some comments, mainly related with tensile test results and interpretation.
Response 1: Dear reviewer, thank you for the review of revised manuscript and your suggestions, which can help improve our paper.
Point 2: The use of a stress-strain curve in experiments (figs 8) has no sense as the sample is made of two different materials with two different cross sections. Authors note that they use the smallest cross-section for deducing the stress, but it has no scientific meaning. They must use load-displacement results. Getting area reduction or elongation on this samples has no meaning, well you can measure it but you cannot compare it directly with the base material. Figure 8 now include the stress-strain curves of base materials, In these cases the thinnest of each welded sample. As previously mentioned, load-displacement graphs have more sense. It will be interesting to add the load-disp curves for both plates on each case, then in a single figure we will have the welded sample, the material A and the material B together.
Response 2: We revise Fig. 8 and Fig.9. as suggested reviewer. In a single figure, we add the curves of the welded sample, the material A and the material B together. We no longer have “load-displacement” curves.
Point 3: Figure 15 show some EDAX results but authors do not comment anything about these results on the text. Take advantage of those measures and analyse them.
Response 3: The advantage of the EDX analysis, in this case, is exact identification of the chemical composition of each compound in the pore. This will contribute to the search for the cause of the pore formation.
Porosity in the TWB´s could be due to various reasons e.g. keyhole instability, improper gas shielding, surface contamination, improper edge preparation; and improper setting of welding parameters.
Point 4: Conclusion 3 is related with tensile test results. Direct comparison between welded samples and base material is not possible. If you want to conclude something you must make a more intense study deducing mechanical properties of and ideally welded sample, i.e. taking into account both sections, sample geometry, welding location and mechanical properties of welded plates. Without this study conclusion III is not valid.
Response 4: Yes, that´s the true. Conclusion iii) was revised as follows:
“A change of the mechanical properties of the welded joints and the base materials of the TWB was observed. The decrease of the ductility of both the TWB1 and TWB2 can be related to the heat–affected welding zone due to hardening. The thickness ratio and strength ratio has an effect on failure whereby the weaker material (low thickness, lower mechanical properties) deforms more and fails first. A larger thickness ratio forces more deformation into the weaker material and the strain is concentrated there, which results in premature failure.”
Thank you again for a detailed review.

This manuscript is a resubmission of an earlier submission. The following is a list of the peer review reports and author responses from that submission.
Round 1
Reviewer 1 Report
The topic of the research is interesting and surely falls within the scope of the Journal, nevertheless the reviewer has the following major concerns:
- It is clearly stated in the abstract and in the introduction that the effect of laser welding parameters was to be investigated, but this was not the case. The authors just limited their study to two TWBs realized using a single set of process parameters each.
- The state of the art in the introduction is not exhaustive, with references often partial, particularly when used to reference general techniques and procedures rather than specific zinc coated laser welding.
- The analysis and the presentation of the results of the mechanical tests were sometimes poor and not rigorous. Some of the considerations put forth by the authors were not fully correct.
Moreover, the English level and style should be strongly revised to correct grammar errors/typos and improve the overall readability of the paper.
For these reasons, the reviewer does not recommend the publication of the paper in Metals in its present form.
Detailed comments are as follows:
Introduction
- The overall quality of the writing makes some sentences difficult to read and understand.
- The references [1-4], [7-9] and others are focused on zinc coated sheet welding only, while they should be referencing general laser weldment literature works, or papers related to TWBs realization.
- Lines 43-48. The same concept is repeated twice.
- Last lines: It is not true that experiments are carried on a wide range of materials and combinations. Just two combinations (TWB1 and TWB2) have been studied.
Section 2.1.
- The mechanical performance of the base materials should be quantified and indicated in the paper. Stating that one has “superior” properties is too vague.
Section 2.2.
- Few details are provided on the equipment used for the actual weldment of the TWBs.
- Only one set of parameters is used for each welded TWB coupon, as reported in table 3. Thus, it is not correct to state that the authors studied the effect of joining parameters and laser settings. The work just investigates the results of a unique parameter choice on the final performance of the two CO2 Laser weldments.
- No indications on the dimensions of the TWB coupons from which specimens are cut are provided.
Section 2.4.
- Line 117-118:what does “controlled by computer under test load” mean?
- Line 119: percent over all elongation → elongation at break
- Line 120-121: the definition of a stress-strain diagram for the TWB is not appropriate.The specimen is made of different “materials”: base alloys, HAZs, FZ, and additionally its parts have different thicknesses. It would be better to use a global force-displacement diagram. A normalization is possible but the term stress-strain curve is misleading, since in this case it does not refer to a single material.
Section 3.3.
- Line 210: Figure 8 → figure 10.
- Tables might be used in place of Figures 8a, 9a.
- Figures 8b, 9b should be replaced with diagrams presenting a more correct information (see the relative comment in section 2.4.).
- The motivation that the elongation at break of the TWB specimen is lower than the one of the base material specimens is due to the fact that just one part gets highly plastically deformed and not the whole specimen. The reduced deformation in the transversal direction has a limited effect on the overall axial elongation, instead.
- Line 220. This sentence is not clear.
- Lines 223-238: here the same concept (the thinner part of the TWB gets deformed and the thicker behaves rigidly) is repeated more than one time.
- Line 232: the assumption that the higher microhardness of the weld seam with respect to the hazs and base materials shifts the fracture away from the weldment is not always true.
- The explanation given to justify the presence of defects observed in one welded joint is not satisfactory. It is a mere enumeration of all possible causes, without any supporting evidence.
- In the last lines before the concluding section it is stated that TWB2 exhibits a good formability, but the comparison of data in Figure 11 and 17 shows that TWB1 performs better, though presenting defects in the weldment.
Reviewer 2 Report
Dear Authors,
This work is a good study work but does not add any news in terms of Industrial or science applicarion.
I find this points of weak in your Work:
1- In the introductione you speak about:
"The experimentalanalysis have been done to study the effect of parameters of laser welding"
but no Design of Experiment and correlation beetwen shape, microstructure, and in general about the quality of the welds is reported;
2- You affirm that: "The novelty of the paper concerns the use of such combinations of materials that rarely occur in the literature.
Of course, but you do not give any industrial possible application for justify the choice of TW1 and TW2 type of joints;
3- The CO2 was a glourious kind of laser but in 2018 appears an obsolete technology. Could you, for example, report some present-day application of CO2 laser in automotive Industry ( FCA, VW, PSA.. etc..) I think all the Industry are now using Disk or Fiber laser;
4- You had performed tensile test on a transversal Butt Joint and evaluate Ry and A%. Thi is completely wrong. Even if you have a standard Butt joint ( same material and same thickness) in a transversal tensile test you can evaluate just the RM. In a transversal tensile test of a butt weld you test 3 different materials in series ( BM+HAZ+FZ) so, you cannot evaluate Ry or A%.
Some scientific approach use local Analisys of Image (DIC) to evaluate the evolution of local deformation in function of the materials (HAZ, FZ).
Only in omogeneus material you can evaluate the Ry and A% see ( ISO 4136 you can record just Fm, Rm and location of fracture)
About Hardness test you speack about load velocit [mN/s] but the standard prescribe usually time of hold indentation time ( 10-15 s).
It is not clear the repetition and the localizzation of indentation ( the X axe of the graph report a Number o VTP not a distance )
The conclusions are more or less obvious.
Reviewer 3 Report
The work presented in this paper revolves around mechanical characterization of TWB. While the topic merit extensive research due to its technological importance, I did not find this paper to contribute much to the field. Given below is a short list of additional data needed for the paper to merit publication in metals. Moreover, some conclusions, which are not general and well known are needed to make the paper acceptable.
The microstructure images are at a low quality and not much can be learned from them.
In general, more attention to the microstructural characterization is needed.
The authors have conducted tensile measurements on TWB specimens. I would have liked the stress-strain curve of the two base metals to be presented together with the TWB.
The stress-strain curves of the TWB doesn't provide much information. According to the authors only the thinner section deforms. If this is indeed the case then the curve should coincide with the curve of the base material. no such comparison was given.
Instead of assuming the thickness to be that of the thinner section, the stress gradients in the TWB could have been calculated. along with a DIC measurement of the strains a details description of the strain partitioning and stress evolving in the TWB could have been presented.
As for the erichsen test, some of the text referring to those is more speculative the relying on actual observations. it does not belong in a result section but rather as a discussion and can be supported by calculations.
As stated by the authors the quality of welding was poor. I would have expected them to characterize in details the weld and suggest something regarding the density of flows. After all, we do not know if all the specimens tested contained flow or not.
Later on the first conclusion is that the weldability is good. I don't understand what this claim is based upon. especially when the micrographs suggests large pores.
Also, I am missing statistics and error bars for the experimental results.
Finally the conclusions in the paper does not provide any new information to the community. Also it is not always clear if this is a conclusion or hypothesis. e.g. in conclusion iii it is stated :"The decrease of the ductility of the base material for both the TWB1 and TWB2 is 293 related to the heat-affected welding zone due to hardening" other then the fact that this is many times the case, what clear evidence is given?
Similar to the 10 can be said regarding conclusion 5.
other conclusions are simply restating the experimental observation, while other give very general guide lines such as use clean surfaces when welding and consistent weld surface finishing.
To conclude, I think that major revisions are required
Reviewer 4 Report
Paper is interesting, but it could be improved in several ways.
In section 2.1 first paragraph in lines 86-89 they do not say that they are referring to TWB1, but in line 90 they comment that that sentence is related to TWB2. They should include the reference to TWB1 in first sentences.
Table 2 is wrong as is related with TWB2 and in titles shows “Chemical composition TWB1 [%]”.
Table 3 mm.s-1 the “dot” symbol must be “·” instead of “.”.
In several places a double empty space were used after dots “.”. The paper must be uniform. Use double empty spaces before all “.”, or use a single space before each “.”. But do not mix both criteria.
Lines 117-118. It is not clear what “under tensile load” means. The computer is controlling the machine load instead of the displacement? Make it clear.
Lines 133. “An overall photo of the microstructure…” it is the cross section, so I think it is better to write: “An overall cross section photo of the….”
Figure 2. Draw a line on images indicating the “Specimen direction”.
Along the test authors use “top” and “root” zones of the weld. I guess that they mean “top” and “bottom”. Clarify this please.
Line 142: The microstructure was also shown in figure 2. Figure 3 shows “More detailed microstructures…”.
Line 148. The hole was “observed” not “identified”.
Line 156. With polyedric grains means “equiaxial” ones?
Figure 4 must be rotated 90º and specimen direction should be included. If you do not rotate the terms “top” and “bottom” has no sense in lines 160…
Line 185: You mane the TWB1a sample. What that a means? It always look that you only used a sample for each material, but here you add and “a” and in figure 10 there are three samples.
Figure 6: What is the distance between “Vickers test points”? this parameter is interesting as it will help to determina the real size of the well and the HAZ.
Figure 8 and 9 will be better if you use just vertical lines or bars without “3D effects”. In case you print your papper in a laser printer it is really difficult to distinguish between both materials. Try making one color clearer and the other darker. It will also be interesting to use “2 vertical axis”. One on the left in “MPa” for yield stress and tensile strength and the other on the right for the “Elongation [%]”. These figures also show Engineering Stress and Eng Strain… please note it.
Paragraph between lines 217 and 227 is duplicated in lines 228-238. Please erase the first one, as the second includes the references.
Line 230 you propose a decrease in the elongation due to the hardening of the welded zone, but what about the rest of the sample. Now your initial sample is made of only one material but the second ones TWB’s are made of two materials, one thicker that the other. This will make the elongation to be smaller.
Figure 12 is not clear. Clarify what the left imagen and the right ones defines as they are used in different parameters for explaining different observations. This is very confusing. The figure footer must be self-explaining…and a single short sentence is not enough in this case.
Figure 15 notes that the (c) edax corresponds to the base material… but it has a lot of Zn… is this right? I will expect the Zn inside the pore surface, as you theorised in the text. Check the order of this EDAX carefully. Also arrows that connect EDAXs to micrograph zones will help a lot.
Conclusion III is also discusable as the decrease in elongation could be due to the existence of a thicker part on the welded sample.